# Transcriptome Analysis of Human Endogenous Retroviruses at Locus-Specific Resolution in Non-Small Cell Lung Cancer

**DOI:** 10.3390/cancers14184433

**Published:** 2022-09-13

**Authors:** Alessandro La Ferlita, Rosario Distefano, Salvatore Alaimo, Joal D. Beane, Alfredo Ferro, Carlo M. Croce, Philip N. Tsichlis, Alfredo Pulvirenti, Giovanni Nigita

**Affiliations:** 1Department of Cancer Biology and Genetics, The James Comphrensive Cancer Center, The Ohio State University, Columbus, OH 43210, USA; 2Bioinformatics Unit, Department of Clinical and Experimental Medicine, University of Catania, 95125 Catania, Italy; 3Department of Surgery, Division of Surgical Oncology, The James Comphrensive Cancer Center, The Ohio State University, Columbus, OH 43210, USA

**Keywords:** human endogenous retroviruses, HERVs, lung cancer, lung adenocarcinoma, lung squamous cell carcinoma, biomarkers, RNA-seq, transcriptome analysis

## Abstract

**Simple Summary:**

Lung cancer is the leading cause of cancer deaths worldwide. Most lung cancer patients are diagnosed with locally advanced or metastatic diseases, and their prognosis is relatively poor, with 5-year survival rates ranging from 4 to 17%. Consequently, the identification of novel diagnostic lung cancer biomarkers remains crucial. Recently, human endogenous retroviruses (HERVs) have been found to be implicated in cancer development and later employed as novel diagnostic and prognostic cancer biomarkers. In this study, we present the first-ever locus-specific analysis of HERV expression in 515 lung adenocarcinoma (LUAD) and 497 lung squamous cell carcinoma (LUSC) patients’ samples from the TCGA repository. In our study, we identified the differentially expressed HERVs in both TCGA-LUAD and TCGA-LUSC cohorts, we examined their impact on signaling pathways using in silico models, and we described HERVs’ association with overall survival (OS) and relapse-free survival (RFS).

**Abstract:**

Lung cancer is the second most commonly diagnosed cancer and the leading cause of cancer deaths worldwide. Among its subtypes, lung adenocarcinoma (LUAD) and lung squamous cell carcinoma (LUSC) are the most common, accounting for more than 85% of lung cancer diagnoses. Despite the incredible efforts and recent advances in lung cancer treatments, patients affected by this condition still have a poor prognosis. Therefore, novel diagnostic biomarkers are needed. Recently, a class of transposable elements called human endogenous retroviruses (HERVs) has been found to be implicated in cancer development and later employed as novel biomarkers for several tumor types. In this study, we first ever characterized the expression of HERVs at genomic locus-specific resolution in both LUAD and LUSC cohorts available in The Cancer Genome Atlas (TCGA). Precisely, (i) we profiled the expression of HERVs in TCGA-LUAD and TCGA-LUSC cohorts; (ii) we identified the dysregulated HERVs in both lung cancer subtypes; (iii) we evaluated the impact of the dysregulated HERVs on signaling pathways using neural network-based predictions; and (iv) we assessed their association with overall survival (OS) and relapse-free survival (RFS). In conclusion, we believe this study may help elucidate another layer of dysregulation that occurs in lung cancer involving HERVs, paving the way for identifying novel lung cancer biomarkers.

## 1. Introduction

Lung cancer, a malignancy that originates in the epithelium of the respiratory tracts, is the second most commonly diagnosed cancer and the leading cause of cancer deaths worldwide [1,2]. Traditionally, lung cancer is divided into two main groups: small cell lung cancer (SCLC) and non-small cell lung cancer (NSCLC) [3]. The NSCLC represents the most common lung cancer form, which comprises three histological subtypes: adenocarcinoma (LUAD) (~50%), squamous cell carcinoma (LUSC) (~35%), and large cell carcinoma (~15%) [4], with LUAD and LUSC accounting for more than 85% of NSCLC diagnosis worldwide. A lung cancer diagnosis is mainly carried out through imaging technologies and pathological examination. Both strategies present limitations in terms of sensitivity, with only 10–15% of new cases being diagnosed at an early clinical stage [5,6,7]. Hence, most lung cancer patients are diagnosed with locally advanced or metastatic diseases, and their prognosis is relatively poor, with 5-year survival rates ranging from 4 to 17% depending on the stage and metastasis location [8,9]. Consequently, the identification of novel diagnostic biomarkers remains crucial.

Recently, human endogenous retroviruses (HERVs), a class of transposable elements, have been found to be implicated in cancer development and later employed as novel diagnostic, prognostic, and treatment response biomarkers for several tumor types [10,11,12,13,14,15]. In more detail, HERVs are remnants of exogenous retroviruses integrated into the human genome during evolution, accounting for 8% of the human genome. These endogenized forms of viral sequences become established after germ cell infections by exogenous retroviruses. Once successfully integrated into the germline genome, proviruses are transmitted vertically by standard Mendelian inheritance [16]. Indeed, HERVs share genomic similarities to other exogenous retroviruses whose genomes typically consist of a common set of at least four genes: (i) *gag*, which encodes structural matrix and capsid proteins; (ii) *pro*, which encodes the viral protease; (iii) *pol*, which encodes the retroviral enzymes reverse transcriptase and integrase; and (iv) *env*, encoding the viral envelope glycoproteins. Due to accumulated mutations, HERVs have preserved the features of their original provirus to a highly variable extent, ranging from the retention of a complete set of long terminal repeats (LTRs) and retroviral genes to the retention of only fragments of the parental viral genomes [17].

Based on their sequences, HERVs are usually classified into 12 major families [18]. Precisely, their nomenclature refers to the single-letter code of the amino acid carried by the tRNA that binds the tRNA binding site of the viral RNA of that given family used initially as a primer to start the reverse transcription. Among them, the HERV-K family, which harbors the lysine tRNA binding site, is the most recently acquired by humans around three million years ago, and it is commonly subdivided into 11 subgroups (HML-1 through HML-11) [18]. Due to their relatively recent integration into the human genome, members of the HERV-K subgroup, called HML-2, still contain genes with intact open reading frame (ORF) that can encode for retroviral proteins [10]. Moreover, the expression of such elements may be deregulated in several human cancers. However, the exact roles of these genes in cancer development and progression remain unknown [10]. For this reason, HERV-K HML-2 elements are attracting significant interest in cancer research [10]. However, it is not only the HERV-K elements that might be relevant for cancer development. Emerging evidence has shown that several families of HERVs with highly mutated inactive protein-encoding genes are still actively transcribed, producing HERV-derived non-coding RNAs (ncRNAs), whose function and impact on cancer development remain to be elucidated [16]. In addition, the LTRs present at the 5’ and 3’ extremities of the HERVs may recruit transcription factors enhancing the transcription of host cell genes located in their proximity which may, in turn, affect the gene expression and, therefore, the biology of the cancer cells [19]. Recent studies have also shown that HERVs may regulate the host immune response to cancer cells by several mechanisms: (i) by inducing a mimetic state of a viral infection; (ii) by generating tumor-specific antigens; or (iii) by inducing the expression of genes associated with immune response [18]. Consequently, their potential importance in immunotherapy raised significant interest in HERVs in recent years [18].

Although important cancer initiatives such as The Cancer Genome Atlas (TCGA) [20] have released an enormous amount of RNA-seq data potentially useful for the characterization of HERV expression in several human cancers, very few studies [21,22,23,24,25,26,27] have examined HERV expression in the TCGA cohorts. In addition, some of these studies [24,25,26,27] leveraged pipelines not explicitly designed for HERV characterization and quantification, lacking the resolving power to quantify HERVs at the genomic locus level. Indeed, quantifying HERVs with standard short-read RNA-seq technologies is challenging due to the repetitive nature of such elements and the consequent uncertainty in fragment assignment because of sequence similarity. Therefore, specific bioinformatics pipelines must be used to address this task. Several approaches have been proposed that account for read mapping uncertainty using statistical models. Among them, Telescope [28] accurately estimates HERV expression at the locus-specific level. Telescope addresses uncertainty in fragment assignment by reassigning ambiguously mapped fragments to the most probable source transcript leveraging a Bayesian statistical model [28]. Benchmark analyses performed by the Telescope’s authors showed that their tool outperforms other methods for HERV quantification, providing the highest resolution since it estimates their expression at the genomic locus level rather than at the HERV subfamily level [28]. Recently, Telescope was also successfully used to identify dysregulated HERVs in head-neck, prostate, breast, colon, and uveal melanoma cohorts from TCGA [21,22,23]. However, other common tumor types such as lung cancer are surprisingly still overlooked. In the present study, we developed an analysis workflow that relies on Telescope [28] for HERV quantification and analysis. For the first time, we performed a locus-specific characterization of the HERV expression in TCGA-LUAD and TCGA-LUSC cohorts, identifying the potential consequences of their deregulation in NSCLC. Specifically, (i) we profiled the expression of HERVs in TCGA-LUAD and TCGA-LUSC cohorts; (ii) we identified the dysregulated HERVs in both lung cancer types; (iii) we evaluated the impact of the dysregulated HERVs on signaling pathways using neural network-based predictions; and (iv) we assessed the association of HERV expression with overall survival (OS) and relapse-free survival (RFS).

## 2. Materials and Methods

### 2.1. Pre-Processing of the RNA-Seq Data

Paired-end RNA-seq data (Illumina) were downloaded in BAM format (.bam) from TCGA-LUAD and TCGA-LUSC cohorts using the Genomic Data Commons (GDC) data transfer tool [29]. We tested the read strandedness using the *how_are_we_stranded_here* Python library [30], retaining only unstranded samples. Of the two cohorts, only TCGA-LUAD presented 22 stranded samples out of 598 (576 retained), while all TCGA-LUSC samples (545) were unstranded. Only primary tumor (non-metastatic) and solid normal tissue (non-cancerous tissues adjacent to the tumor) samples were employed for downstream analyses. Downloaded BAM files were first converted to FASTQ format (.fq) using BioBamBam2 (bamtofastq function) and then trimmed using Trim Galore (https://www.bioinformatics.babraham.ac.uk/projects/trim_galore/ (accessed on 29 April 2022)), which includes FASTQC [31] and Cutadapt [32]. Trimmed reads were then remapped to the human genome (HG38 assembly) using Bowtie2 [33], allowing up to 100 alignments per read (-k 100). Afterward, the mapped reads in SAM format were converted into BAM format, sorted (for coordinates), and indexed using Samtools [34]. Finally, HERV quantification was performed using Telescope [28] with its GTF annotation file. All the pre-processing analyses were performed using the Ohio supercomputer (OSC). A schematic representation of the data pre-processing together with the complete analysis workflow is shown in Appendix A. The generated raw read count matrices were then used as input for all the downstream analyses presented in this study (see following sections). Raw count matrices of HERVs across all analyzed samples for TCGA-LUAD and TCGA-LUSC cohorts can be found in Appendix A, respectively.

### 2.2. Differential HERV Expression Analysis

To perform the differential HERV expression (DE) analysis, we first normalized the raw read counts via trimmed mean of M values (TMM) using edgeR [35] and then filtered out low expressed HERVs, whose mean value was less than five across all samples. Afterward, the raw read counts of the retained HERVs was log2-transformed leveraging *Voom* and then used for the DE analysis, leveraging the Limma R package [36]. Two different DE analyses were performed. In the first DE analysis, we considered only those tumor samples with matched adjacent non-cancerous tissues taken from the same patient (paired analysis). In particular, we compared 58 tumor/normal samples in TCGA-LUAD and 47 tumor/normal samples in TCGA-LUSC. Instead, in the second DE analysis, we compared the tumor samples at the early stages (IA) vs. the tumor samples at the advanced stages (III and IV) for both TCGA-LUAD and TCGA-LUSC. Notably, not all tumor samples available for both TCGA-LUAD and TCGA-LUSC cohorts have information about their clinical stages. Therefore, only the tumor samples whose stage (IA, III, or IV) was reported were taken into consideration (244 Samples for TCGA-LUAD and 171 samples for TCGA-LUSC). In both analyses, only HERVs with a |Log_2_FC| > 0.58 (|Linear FC| > 1.5) and an adjusted p-value (Benjamini–Hochberg correction) < 0.05 were considered differentially expressed. A schematic representation of the DE analysis and the complete analysis workflow is shown in Appendix A. The plots generated for showing the results of the differential HERV expression analysis, such as volcano plots, Venn diagrams, heatmaps, and circos plots, have been drawn using EnhancedVolcano [37], InteractiVenn [38], pheatmap (https://cran.r-project.org/web/packages/pheatmap/index.html (accessed on 16 May 2022)), and circlize [39], respectively.

### 2.3. Differential Gene Expression Analysis

The gene raw read counts (coding and non-coding RNAs) for the TCGA-LUAD and TCGA-LUSC cohorts were downloaded via the GDC data portal (TCGA v33). Only the tumor samples paired with the adjacent normal tissue samples were used for the differential gene expression analysis. We first normalized the gene raw read counts via TMM and then filtered out low expressed genes whose geometric mean was <1 TMM across all samples. Afterward, the raw read counts of the retained genes were log2-transformed leveraging *Voom* and used for the DE analysis using Limma. Genes with an adjusted p-value (Benjamini–Hochberg correction) < 0.05 were considered differentially expressed.

### 2.4. Correlation Analyses

The differentially expressed HERVs identified from the TCGA-LUAD and TCGA-LUSC cohorts were correlated with the differentially expressed genes (DEGs) in the same TCGA cohorts and also with a list of 13 relevant oncogenes and adaptive immunity regulators used as therapeutic targets for NSCLC treatment [40,41], focusing only on the primary tumor. Normal and metastatic samples were not considered for this analysis. HERV and gene raw read count matrices were normalized using the read per million (RPM) formula to scale the raw library sizes. Subsequently, we correlated the HERV expression (Spearman correlation) first with the DEGs and second with the list of selected oncogenes and adaptive immunity regulators for each cohort. The resulting p-values were adjusted using the Benjamini–Hochberg correction. Only DEGs with an absolute value of Spearman correlation coefficient > 0.2 and an adjusted *p*-value < 0.05 were considered correlated with that specific HERV, while only HERVs correlated with the selected oncogenes and adaptive immunity regulators with an absolute value of Spearman correlation coefficient > 0.35 and an adjusted *p*-value < 0.05 were plotted on the correlation networks generated by Cytoscape (v3.9.1) [42] for both lung cancer cohorts.

### 2.5. Pathway Analysis

To predict the effects of the dysregulated HERVs identified in the TCGA-LUAD and TCGA-LUSC cohorts on metabolic and signaling pathways, we performed a neural network-based topological pathway analysis using MITHrIL [43]. For each differentially expressed HERV, we used the list of its correlated DEGs (identified as described in the previous paragraph) with their Entrez ids and respective Log_2_FC values as an input for MITHrIL. Only DEGs with an absolute value of Spearman correlation coefficient > 0.2 and adjusted *p*-value (Benjamini–Hochberg correction) < 0.05 were considered correlated for that specific HERV and used for the HERV-specific pathway analysis. Hence, single MITHrIL analyses were run for each differentially expressed HERV identified either in TCGA-LUAD or TCGA-LUSC cohorts, and only pathways with a *p*-value < 0.05 were selected for the further clustering analysis. For the latter, we considered each pathway’s “Corrected Accumulator” value generated by MITHrIL. First, a consensus clustering analysis was performed to identify the number of potential clusters for both HERVs and enriched pathways using the ConsensusClusterPlus R package [44]. Afterward, a heatmap was generated using the ComplexHeatmap R package [45], where the rows were the HERVs, and the columns were the enriched pathways. The Spearman correlation distance was employed for both columns and rows. Pathways that were enriched in less than 30% of the differentially expressed HERVs were excluded from the clustering analysis.

### 2.6. Survival Analysis

For the survival analysis we considered all those HERVs with a geometric mean > 1 RPM across all samples. After the filtering step, we normalized the expressed HERVs with the TMM method, and we finally carried out both OS and RFS analyses by employing a univariate Cox regression model leveraging the *CoxPHFitter* function from the *lifelines* (v0.27.0) Python package. HERVs with a BH-adjusted Cox *p*-value < 0.1 were considered significant.

## 3. Results

### 3.1. HERV Profiling in TCGA-LUAD and TCGA-LUSC

To investigate the HERV expression patterns in NSCLC, we analyzed RNA-seq data from the LUAD and LUSC cohorts available in the TCGA repository. In particular, we examined RNA-seq data from 515 and 497 primary tumor tissues for TCGA-LUAD and TCGA-LUSC cohorts, respectively. Normal samples available for both cohorts were not used in this analysis but used in the differential HERV expression analysis (see the following section). A summary of the available clinical information for both TCGA-LUAD and TCGA-LUSC cohorts is reported in Appendix A.

We developed an in-house workflow (described in the Section 2 and Appendix A) that relies on Telescope [28] to accurately detect and quantify HERVs at the genomic locus. Of the 14,174 HERV genomic loci analyzed, 4227 (29.8%) and 4439 (31.3%) were found expressed (average RPM > 5) in TCGA-LUAD and TCGA-LUSC cohorts, respectively. Notably, the two cohorts shared a large set of expressed HERVs (73.3%) (Figure 1A). Across them, the HERV families HERV-H, HERV-K, and MER4 were among the most represented ones. Although these families include more elements than others, they still show a considerably higher percentage of expressed HERVs per family. The lists of the top 10 most represented HERV families for both TCGA-LUAD and TCGA-LUSC are shown in Appendix A.

Afterward, we performed a uniform manifold approximation and projection (UMAP) map reduction analysis (https://cran.r-project.org/web/packages/umap/index.html (accessed on 11 May 2022)) of the top 100 most variable HERVs based on their Median Absolute Deviation (MAD) values. The results showed that LUAD and LUSC samples form two well-separated clusters indicating that these two different histological lung cancer subtypes have different HERV expression patterns (Figure 1B). Moreover, additional UMAP analyses using the top 100 most variable HERVs identified either in LUAD or in LUSC were performed to establish whether tumors of the same molecular subtype cluster together using the HERV expression. The molecular subtype classification for TCGA-LUAD and TCGA-LUSC was retrieved from Chen F. et al. [46], and it takes into consideration the expression and genetic alterations of several genes such as SOX2, PTEN, TP53, KRAS, EGFR, and many others. The analysis showed in LUAD three clusters where tumor samples belonging to the molecular subtypes AD.1, AD.2, AD.3, AD.4, and AD.5a tend to cluster with members of the same molecular subtypes while the other subtype AD.5b present tumor samples that are more heterogeneous in terms of HERV expression (Figure 1C). On the other hand, in LUSC we saw two better-defined clusters, wherein one of these clusters, tumor samples belonging to the molecular subtypes SQ.1 were more represented, while in the other one the molecular subtypes SQ.2a and SQ.2b were predominant (Figure 1D). To see details of the molecular characteristics of each molecular subtype refer to the paper of Chen F. et al. [46].

### 3.2. Dysregulation of HERVs in TCGA-LUAD and TCGA-LUSC

To identify dysregulated HERVs in TCGA-LUAD and TCGA-LUSC cohorts, we performed differential expression analysis comparing the primary tumor samples (non-metastatic) with their adjacent normal tissue samples. For the TCGA-LUAD cohort, 58 tumor samples were compared with their respective 58 adjacent normal tissue samples, while for TCGA-LUSC, 47 tumor samples were compared with their respective 47 adjacent normal tissue samples. Major dysregulations in HERV expression were observed in both cohorts. Specifically, a total of 1522 and 2421 HERVs were found dysregulated (|Log_2_FC| > 0.58 and FDR < 0.05) in the TCGA-LUAD and TCGA-LUSC cohorts, respectively (Figure 2A,B). Dysregulated HERVs were found dispersed throughout the genome (Figure 2C,D). 

We observed a slight predominance of down-regulated (820) HERVs over up-regulated (702) ones in TCGA-LUAD, while TCGA-LUSC showed an opposite trend (1435 up-regulated and 986 down-regulated). Only 27.6% of the up-regulated HERVs (Figure 3A) and 55.1% of the down-regulated (Figure 3B) HERVs overlapped between the two cohorts, suggesting different dysregulation mechanisms in the two lung cancer types. Furthermore, we built two heatmaps, one for TCGA-LUAD and one for TCGA-LUSC samples, using the normalized counts of the identified differentially expressed HERVs. The results for both cohorts revealed two well-defined clusters, one for tumors and one for normal samples (Figure 3C,D). The differential HERV expression analysis results for LUAD and LUSC can be found in Appendix A.

Moreover, several HERV families were found to be dysregulated in both LUAD and LUSC. Families with a higher percentage of differentially expressed HERVs included HERV-H, HERV-K, and MER4. Interestingly, up-regulation of the HERV-H and HERV-K families has been proposed as novel diagnostic biomarkers [47,48,49], while the up-regulation of the MER4 family has been linked to progression-free survival (PFS) and overall survival (OS) [11]. MER4 HERVs have also been proposed as novel biomarkers for the response to immune checkpoint inhibitors (ICI), which, currently, are becoming the standard first-line treatment of advanced NSCLC for patients with high PD-L1 expression [11]. The lists of the top 10 HERV families dysregulated in LUAD and LUSC cohorts are shown in Appendix A.

Finally, an additional differential expression analysis was performed comparing the early stages of NSCLC (IA) against the advanced stages (III and IV). The analysis, with the exception of TCGA-LUAD, did not show remarkable differences in HERV expression. Precisely, in TCGA-LUAD we identified 172 differentially expressed HERVs (|Log2FC| > 0.58 and FDR < 0.05), while in TCGA-LUSC, no strong significant results were detected. These results seem to suggest that the tumor stages clinically determined do not necessarily reflect a gradually increased dysregulation in HERV expression. The results of this analysis are reported in the Appendix A.

### 3.3. HERV-Specific Neural Network-Based Pathway Analysis

To predict the effects of the HERV dysregulation on signaling pathways in the TCGA-LUAD and TCGA-LUSC cohorts, we employed MITHrIL [43] to perform a neural network-based topological pathway analysis focusing on genes whose expression correlates with HERV expression. First, we used a Spearman correlation formula to determine the correlation of the differentially expressed HERVs in either TCGA-LUAD or TCGA-LUSC cohorts with the DEGs identified in the respective cohort (see Materials and Methods section). The list of DEGs whose expression correlates with a given HERV was used as input for MITHrIL to identify the signaling pathways potentially impacted by the given HERV. Subsequently, clustering analysis based on the “Corrected Accumulator” values, which are an indicator of pathway activity generated by MITHrIL (a positive value indicates the up-regulation of a pathway while a negative value indicates the downregulation of a pathway), was performed to identify groups of HERVs that potentially impact pathways in a similar fashion (see Materials and Methods section). A schematic representation of the pathway analysis together with the complete analysis workflow is presented in Appendix A.

The results of the HERV-specific pathways analysis showed that several cancer-related signaling pathways were enriched in TCGA-LUAD and TCGA-LUSC cohorts (Figure 4A,B). Relevant examples include the MAPK signaling pathway, the mTOR signaling pathway, the RAS signaling pathway, the PI3K-Akt signaling pathway, the ErbB signaling pathway, the cGMP-PKG signaling pathway, the HIF-1 signaling pathway, and the cAMP signaling pathway (Figure 4A,B) suggesting that HERVs may be involved in important signaling pathways known as critical in cancer development and progression. Moreover, also several pathways that contribute to adaptive immunity were enriched in LUAD and LUSC cohorts. Relevant examples include the T cell receptor signaling pathway, the B cell receptor signaling pathway, and natural killer cell-mediated cytotoxicity (Figure 4A,B).

Importantly, the HERV profile correlations with enriched signaling pathways identified by MITHrIL in LUAD and LUSC exhibited significant differences. This observation reflects the fact that only 27.6% of up-regulated and 55.1% of down-regulated HERVs were in common between the two cohorts. The pathway analysis results reporting the corrected accumulators generated by MITHrIL for both TCGA-LUAD and TCGA-LUSC can be found in Appendix A.

In conclusion, the clustering analysis on the HERV-specific pathway analysis results identified four main clusters of HERVs in TCGA-LUAD and three main clusters of HERVs in TCGA-LUSC, in which within the clusters, HERVs seem to impact the enriched pathways similarly (the list of HERVs for each of these clusters in both TCGA-LUAD and TCGA-LUSC can be found in Appendix A). Moreover, in TCGA-LUAD, we observed a higher percentage of HERVs belonging to the HERV-H, HERV-K, and MER4 families in the fourth cluster compared to the other clusters, while in TCGA-LUSC, we observed a higher percentage of HERVs belonging to the HERV-H family in the second cluster, and HERV-K in the first cluster. On the other hand, the other HERV families were more similarly distributed along the clusters. Appendix A show the top 10 dysregulated HERV-families for each cluster identified in TCGA-LUAD and TCGA-LUSC cohorts, respectively.

### 3.4. Correlation of HERVs with Oncogenes and Adaptive Immunity Regulators in TCGA-LUAD and TCGA-LUSC

The pathway analysis (described in the previous paragraph) showed enrichment of several signaling pathways notoriously known to be relevant in cancer progression and tumor immunity indicating a possible regulation of them by the HERVs identified as differentially expressed in TCGA-LUAD and TCGA-LUSC. To investigate more, we selected some genes involved in the regulation of such pathways in order to see their connections with the HERVs identified as differentially expressed. Precisely, we selected ten oncogenes and three adaptive immunity regulators used in the clinic as therapeutic targets for NSCLC treatment [40,41] and we built two correlation networks (one for LUAD and one for LUSC, respectively) showing their connections with the HERVs.

The analysis showed several HERVs to be statistically significantly correlated with the selected genes. In more detail, as can be observed from the correlation networks shown in Figure 5A,B for TCGA-LUAD and TCGA-LUSC, respectively, we identified many HERVs to be positively correlated with the selected genes (red edges) and very few negatively correlated (blue edges). For graphic reasons, only the HERVs highly correlated with the selected genes (|Spearman correlation coefficient| > 0.35 and FDR < 0.05) were plotted in the correlation networks (some of the selected genes are not reported in the correlation networks because they did not present correlated HERVs over the cutoff). Most of the genes presented a variable number of HERVs that exclusively correlated with their expression. Interestingly, ROS1 presented the biggest cluster for both TCGA-LUAD and TCGA-LUSC (Figure 5A,B), where 100 HERVs were exclusively correlated with its expression in LUAD, 90 HERVs in LUSC, of which 38 HERVs were in common between the two cohorts.

Notably, PDCD1 (PD-1), CD274 (PD-L1) and CTLA4, used as targets of immune-checkpoints inhibitors, had a considerable number of HERVs to be positively correlated with at least two of them (Figure 5A,B). Precisely, we identified 44 HERVs correlated with the aforementioned genes in LUAD, 66 HERVs in LUSC, of which 29 HERVs were in common between the two lung cancer histological subtypes. These results seem to suggest a possible involvement of these elements in the regulation of relevant immune signaling pathways and, potentially, the response to immune-checkpoints inhibitors.

Remarkably, even NTRK1, which has been recently associated to promote resistance to PD-1 inhibitors [50], showed several HERVs correlated with its expression that were also correlated with PDCD1 (PD-1), CD274 (PD-L1) expression in both LUAD and LUSC (Figure 5A,B) supporting the evidence that NTRK1 is involved in immune-checkpoints inhibitors response and that HERVs may play a role in orchestrating such phenomenon. The complete results of the correlation analyses can be found in the Appendix A while the correlation networks can be interactively explored by importing the Appendix A on Cytoscape [42].

### 3.5. Association of HERVs with Survival in TCGA-LUAD and TCGA-LUSC

To assess the relationship between HERV expression and patients’ survival in TCGA-LUAD and TCGA-LUSC cohorts, we conducted overall survival (OS) and relapse-free survival (RFS) analyses using a Cox proportional hazard regression model (see Section 2). Interestingly, we identified 42 and 14 HERVs associated (Benjamini–Hochberg adjusted Cox *p*-value < 0.1) with poor OS and RFS in TCGA-LUAD samples, respectively (Figure 6). At the same time, we characterized 5 and 24 HERVs associated with poor OS and RFS in TCGA-LUSC samples, respectively (Figure 6). 

Significantly, most (89.3% in LUAD and 100% in LUSC) of potential prognostic HERVs have a hazard ratio greater than 1, indicating that their high expression is associated with a poor prognosis. Moreover, 32% of the HERVs associated with a poor OS or RFS in TCGA-LUAD, and 55.2% of the HERVs associated with poor OS or RFS in TCGA-LUSC, were also up-regulated in LUAD and LUSC (compared to their adjacent normal tissues). However, this is the first time that these HERVs have been linked to cancer biology; therefore, no additional information is available in the literature about their role in cancer development.

## 4. Discussion

This study presents a locus-specific analysis of HERV expression in LUAD and LUSC patients’ samples from the TCGA repository. Precisely, we identified the differentially expressed HERVs in both TCGA-LUAD and TCGA-LUSC cohorts, we examined their impact on cell signaling pathways using in silico models, and we assessed HERVs’ association with OS and RFS.

The analysis of transposable elements such as HERVs using standard short-read RNA-seq technologies is challenging because of the repetitive nature of such elements, and the consequent ambiguity in assigning RNA-seq reads to the most probable locus. This problem has been recently addressed with the development of novel methodologies. In particular, Telescope [28] emerged over the other existing methods by allowing an accurate estimation of HERV expression. Telescope uses a Bayesian statistical model to reassign ambiguously mapped fragments to the most probable source gene. Moreover, it estimates HERV expression at the genomic locus level rather than at the HERV subfamily level, providing a higher resolution power than the other methods [28]. Based on these considerations, we design an in-house workflow that relies on Telescope for quantifying HERV expression from the RNA-seq data of the TCGA-LUAD and TCGA-LUSC cohorts (Appendix A).

Our analysis addressed the genomic mapping of 14,174 HERV loci, of which 4227 (29.8%) were expressed in the TCGA-LUAD cohort, and 4439 (31.3%) were expressed in the TCGA-LUSC cohort. The HERV subfamilies expressed in a higher percentage in TCGA-LUAD and TCGA-LUSC were HERV-H, HERV-K, and MER4 (Appendix A). Although 73% of HERVs were expressed in common in both lung cancer types (Figure 1A), a map reduction analysis of the top 100 most variable HERVs showed that TCGA-LUAD and TCGA-LUSC samples were well-separated in two distinct clusters (Figure 1B). Additional cohort-specific UMAP analyses also showed a certain correlation between HERV expression and LUAD and LUSC molecular subtypes (Figure 1C,D) defined by Chen F. et al. [46]. These take into consideration the expression and genetic alteration of several genes that are relevant in cancer biology [46].

Following the identification of HERVs expressed in TCGA-LUAD and TCGA-LUSC, we questioned which of those HERVs may be up-regulated or down-regulated in tumor samples relative to their adjacent normal tissues. This analysis identified 1522 HERVs dysregulated in TCGA-LUAD and 2421 HERVs dysregulated in TCGA-LUSC (Figure 2A,B). Dysregulated HERVs were distributed throughout the genome (Figure 2C,D). Of the up-regulated HERVs, only 27.6% were up-regulated in both TCGA-LUAD and TCGA-LUSC (Figure 3A); at the same time, of the down-regulated HERVs, only 55.1% were down-regulated in both cohorts (Figure 3B), suggesting that different dysregulation mechanisms involving HERV expression occur in these two different lung cancer types. However, the most frequently dysregulated HERV families, such as HERV-H, HERV-K, and MER4, were the same in both TCGA-LUAD and TCGA-LUSC (Appendix A). Notably, the upregulation of the HERV-H and HERV-K families in lung cancer has been observed by others, and it has been suggested as a potential diagnostic biomarker [47,48,49]. The expression of members of the MER4 family has been associated with differences in progression-free and overall survival. At the same time, it has been proposed as a biomarker for the response to immune checkpoint inhibitors [11].

To determine whether HERV dysregulation and functional characteristics of NSCLC are linked, we first examined the correlation between the expression of individual deregulated HERVs and the expression of DEGs (only protein-coding ones) in both TCGA-LUAD and TCGA-LUSC cohorts. The list of DEGs with a significant Spearman correlation coefficient for a given HERV was then used as input for the MITHrIL functional pathway analysis. MITHrIL [43] is a neural network-based topological pathway analysis algorithm that fully exploits the topological information encoded by the KEGG’s pathways [51] to compute perturbation scores. KEGG’s pathways are modeled in MITHrIL as complex graphs where each node is a biological element (protein-coding gene, miRNA, or metabolite), and each edge is an interaction between nodes. The functional enrichment analysis identified several pathways notoriously known to be crucial in cancer development and progression such as MAPK, PI3K/AKT/mTOR, RAS, ErbB, and HIF-1 signaling pathways as well as pathways involved in adaptive immunity in both TCGA-LUAD and TCGA-LUSC cohorts (Figure 4A,B). Further correlation network analyses between HERV expression and key genes involved in the aforementioned pathways, which are also used as targets in the clinical practice for NSCLC treatment, revealed interesting clusters of HERVs exclusively or commonly correlated with them in both LUAD and LUSC. Significantly, PDCD1 (PD-1), CD274 (PD-L1), and CTLA4, used as targets of immune-checkpoints inhibitors, had a considerable number of HERVs to be positively correlated with at least two of them in both LUAD and LUSC (Figure 5A,B) suggesting a possible involvement of these elements in the regulation of relevant immune signaling pathways and, potentially, the response to immune-checkpoints inhibitors. These results combined seem to point out that the dysregulation of these pathways in NSCLC may also be caused by the observed dysregulation in HERV expression. HERV expression may, in fact, affect tumor cell functions via multiple mechanisms. First, the LTRs of both complete and defective HERVs contain enhancer and promoter elements whose activity may profoundly affect the expression of neighboring genes. If we consider that at least 20% of the human genes are adjacent to HERVs, it becomes clear that changes in the activity of the HERV’s regulatory elements may have profound effects on the biology of the tumor cells. Another mechanism is the expression of proteins, such as Np9 and Rec, which have been reported to interact with and regulate the activity of transcription factors that, in turn, may promote cancer progression [52,53,54]. In addition to these mechanisms that may promote cancer progression, others may have an opposite effect [55,56,57]. A relevant example includes the expression of viral envelope proteins encoded by intact or near intact members of the HERV-K family, particularly members of the HML-2 subtype, which act as neo-antigens, initiating an adaptive immune response against the tumor cells [10,55,58,59]. Although over the last years several studies have shown that HERV expression might activate an adaptive anti-tumor immunity in multiple human cancers, progress in this area is impeded by the lack of detailed information on the specific members of HERV families who are dysregulated in a given type of tumor. However, prominent immunotherapy clinical trial studies in lung cancer such as POPLAR [60] and OAK [61] have recently publicly released the RNA-seq data that can be potentially reanalyzed for assessing HERV expression, giving the opportunity for evaluating their importance in regulating adaptive immunity and affecting the response to immune-checkpoints inhibitors.

HERV expression-associated functional changes in NSCLCs may alter the NSCLC natural history. To address this question, we examined the association of HERV expression with OS and RFS in both TCGA-LUAD and TCGA-LUSC. This analysis identified 42 and 14 HERVs associated with poor OS and RFS in TCGA-LUAD samples, respectively (Figure 6). At the same time, we also identified 5 and 24 HERVs associated with poor OS and RFS in TCGA-LUSC samples, respectively (Figure 5). Significantly, the vast majority of these HERVs showed a hazard ratio greater than 1, indicating that their high expression is related to a worse prognosis. Moreover, 32% of the HERVs associated with a poor OS or RFS in TCGA-LUAD, and 55.2% of the HERVs associated with poor OS or RFS in TCGA-LUSC, were also up-regulated in LUAD and LUSC (compared to their adjacent normal tissues) suggesting a possible involvement of these HERVs in NSCLC progression. Although the link between HERV expression and patient survival may indicate a causative relationship, this connection may be caused by unknown mechanisms, where HERV expression should be viewed as a valuable biomarker. In this case, the cause of the association should be addressed in future studies.

## 5. Conclusions

This study represents the first locus-specific transcriptome analysis of HERVs in lung cancer. We described dysregulations in HERV expression in TCGA-LUAD and TCGA-LUSC; we also identified their potential impact on regulating essential signaling and immune system pathways, which may affect lung cancer development and progression. Finally, we reported several HERVs associated with worse OS and RFS in LUAD and LUSC. In conclusion, we believe this study might pave the way for identifying novel HERV-based biomarkers in lung cancer and suggests that such HERVs require further investigation and validation to reveal their involvement in crucial signaling pathways.

## Figures and Tables

**Figure 1 cancers-14-04433-f001:**
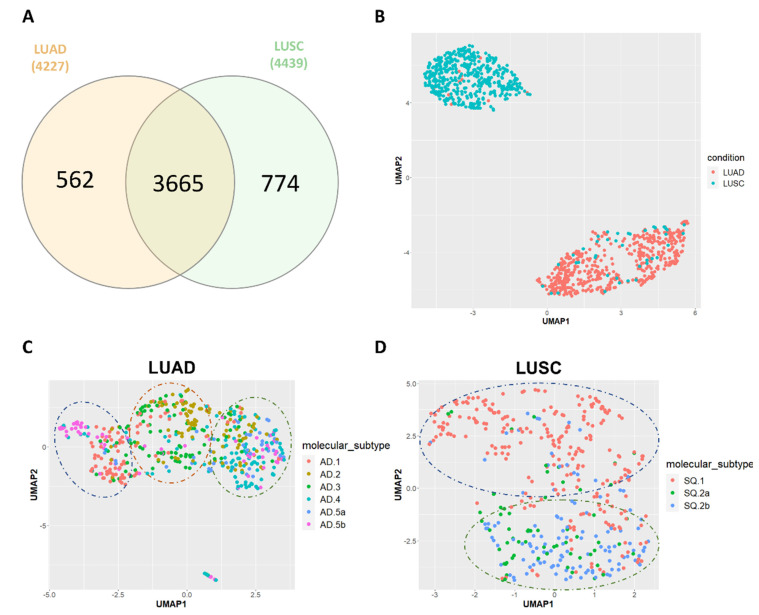
(**A**) Venn diagram showing the number of expressed HERVs identified exclusively in TCGA−LUAD, and TCGA−LUSC, and in common between two cohorts; (**B**) UMAP representation based on the top 100 most variable HERVs showing that TCGA−LUAD (red) and TCGA−LUSC (light blue) sample cohorts form two well-separated different clusters; UMAP representation based on the top 100 most variable HERVs in TCGA−LUAD (**C**) and TCGA−LUSC (**D**), showing the association between HERV expression and molecular subtypes.

**Figure 2 cancers-14-04433-f002:**
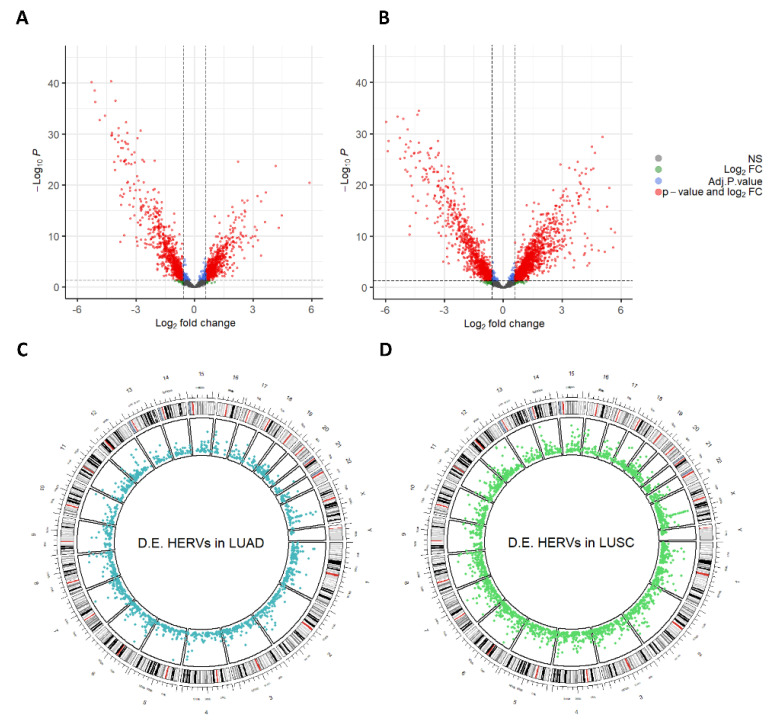
Volcano plots showing the distribution of the differentially expressed HERVs (red) identified in TCGA−LUAD (**A**) and TCGA−LUSC (**B**) in a two-dimensional space where the Y-axis is the significance (−Log_10_P.value) and the X-axis is the magnitude of the change (Log_2_FC); Circos plots showing the distribution of the differentially expressed HERVs identified in TCGA−LUAD (**C**) and TCGA−LUSC (**D**) across all human chromosomes.

**Figure 3 cancers-14-04433-f003:**
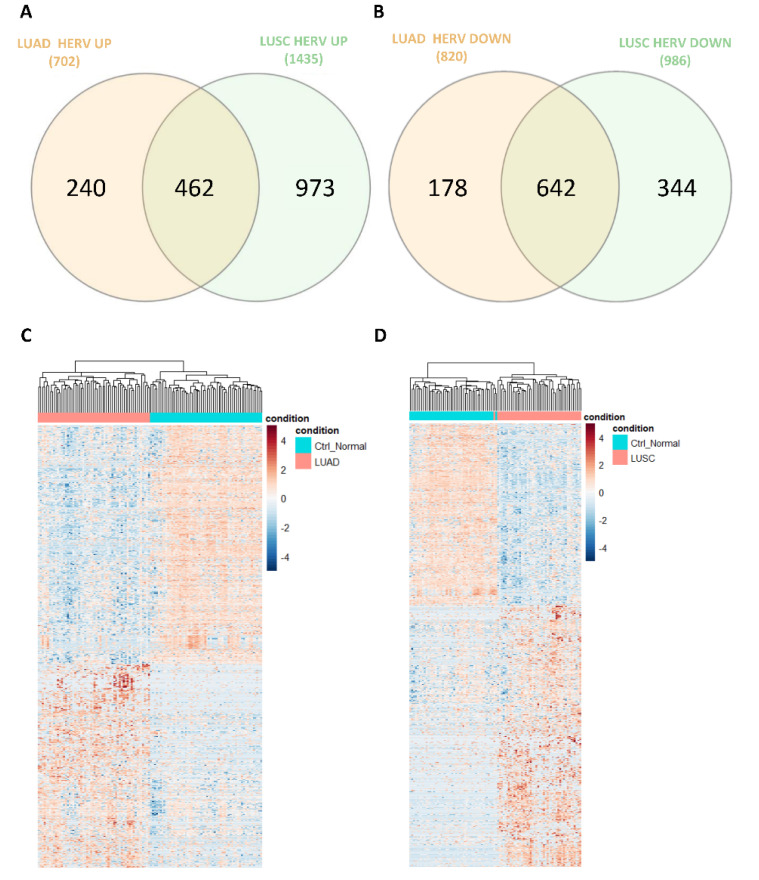
(**A**) Venn diagram showing the number of up-regulated HERVs identified exclusively in TCGA−LUAD, and TCGA−LUSC, and in common between these two cohorts; (**B**) Venn diagram showing the number of down-regulated HERVs identified exclusively in TCGA−LUAD, and TCGA−LUSC, and in common between these two cohorts; heatmap drawn by using the scaled counts (RPM) of the differentially expressed HERVs identified in TCGA−LUAD (**C**) and TCGA−LUSC (**D**) cohorts with their samples clustered accordingly with the Pearson correlation distance.

**Figure 4 cancers-14-04433-f004:**
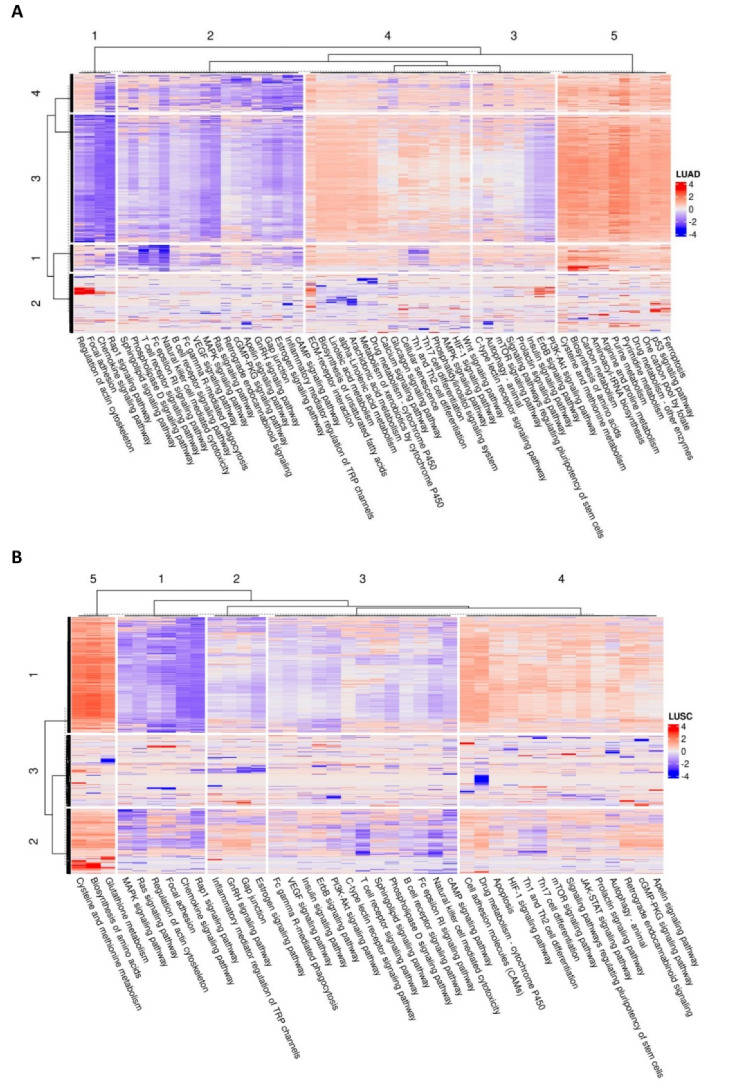
Heatmaps showing the most enriched pathways detected in TCGA−LUAD (**A**) and TCGA−LUSC (**B**) after the HERV−specific pathway analysis performed by MITHrIL. In more detail, the value reported in the heatmap are the corrected accumulators generated by MITHrIL when it computed the perturbation scores, and both columns (pathways) and rows (differentially expressed HERVs) were clustered using the Spearman correlation distance.

**Figure 5 cancers-14-04433-f005:**
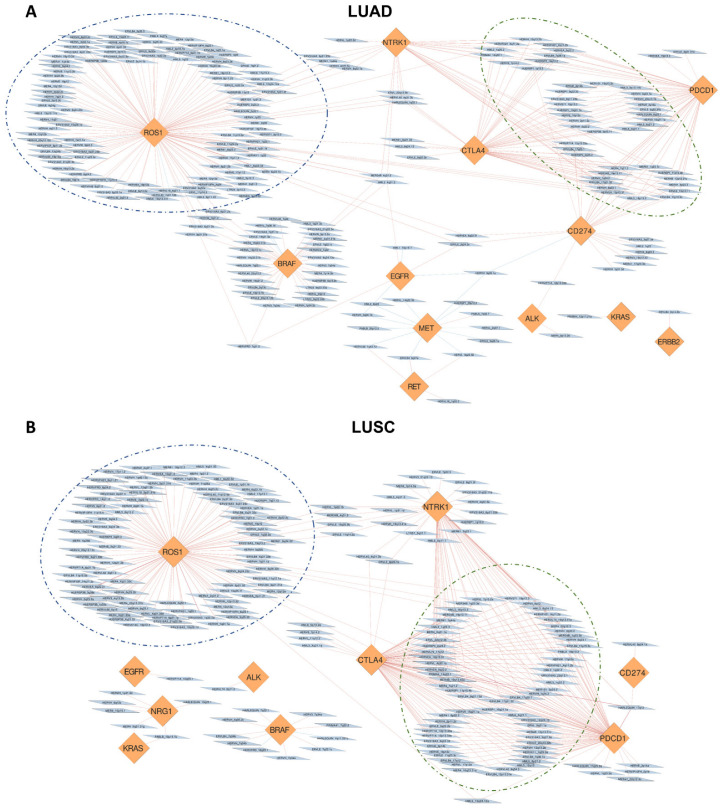
Correlation networks showing the connections between HERVs and a list of selected oncogenes and adaptive immunity regulators in TCGA−LUSC (**A**) and TCGA−LUSC (**B**). In the networks, the blue nodes represent the HERVs and the orange nodes represent the genes.

**Figure 6 cancers-14-04433-f006:**
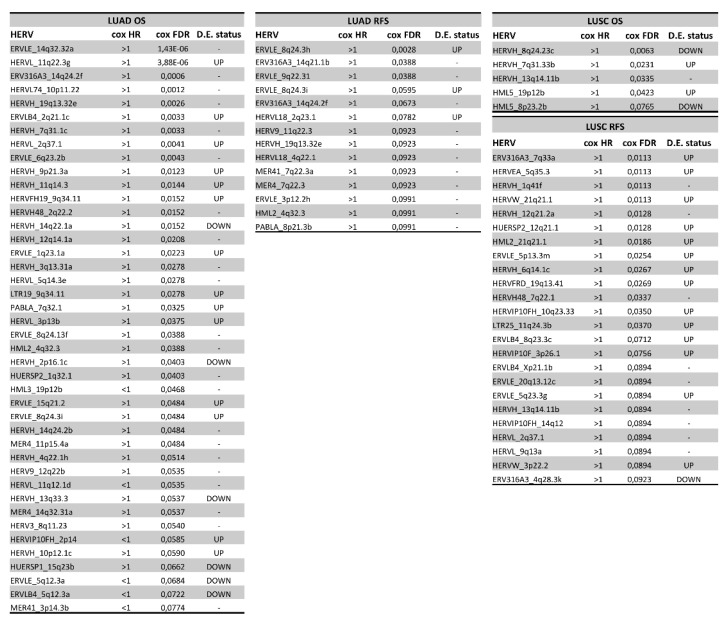
Tables showing the list of HERVs statistically associated with overall survival (OS) and relapse-free survival (RFS) in both TCGA−LUAD and TCGA−LUSC cohorts. In more detail, Cox hazard ratios (HR), and Cox false discovery rates (FDR) values are reported. In addition, it is also reported for each HERV if it has been found up−regulated (up) or down−regulated (down) during the paired tumor vs. normal differential HERV expression analysis.

## Data Availability

Raw sequence data were retrieved via the GDC data portal after obtaining authorization from the data access committee (DBGap Project ID: 11332).

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
