# Peer review of "Transcriptome Analysis of Human Endogenous Retroviruses at Locus-Specific Resolution in Non-Small Cell Lung Cancer"

_cancers, 2022, doi:10.3390/cancers14184433_

Round 1

Reviewer 1 Report

Ferlita et. al. in this manuscript have done transcriptome analysis of publicly available datasets to provide molecular insight into disease outcomes in the context of lung cancer. The overall study is nicely designed and the claims are properly supported by their analysis. The nice part about this article is that the authors scientifically stated their findings and concluded a probable model without overclaiming or overstating.

I think this study can be published as its current form.

Author Response

We are sincerely grateful to the Reviewer for the time and effort dedicated to providing feedback on our manuscript.

Reviewer 2 Report

La Ferlita et al. examines HERVs in two TCGA lung cancer cohorts. The analysis appears sound using a known bioinformatic method. However, there are very little conclusions from these analysis that is of interest to either lung cancer researchers or clinicians. There are numerous analyses that could be done to improve this manuscript. For example, the authors do not break down HERVs by driver oncogene expression or any lung cancer subtypes. Given the importance of immunotherapy in NSCLC, an obvious analysis would be to focus on the relationship between HERVs and immune signaling, which gets a very limited analysis currently. Also, they could analyze RNA-seq data from POPLAR and OAK immunotherapy studies which are now publicly available. In short, this study is not sufficient  in its current form and would need much more analyses to be interesting to the reader. 

Author Response

Reviewer's comment: La Ferlita et al. examines HERVs in two TCGA lung cancer cohorts. The analysis appears sound using a known bioinformatic method. However, there are very little conclusions from these analysis that is of interest to either lung cancer researchers or clinicians.

Authors' response: We are thankful to the Reviewer for putting in efforts to review the paper with the aim of improving our manuscript.

Reviewer's comment: There are numerous analyses that could be done to improve this manuscript. For example, the authors do not break down HERVs by driver oncogene expression or any lung cancer subtypes. Given the importance of immunotherapy in NSCLC, an obvious analysis would be to focus on the relationship between HERVs and immune signaling, which gets a very limited analysis currently. Also, they could analyze RNA-seq data from POPLAR and OAK immunotherapy studies which are now publicly available. In short, this study is not sufficient  in its current form and would need much more analyses to be interesting to the reader.

Authors' response: In the revised version of the manuscript, we performed a new UMAP reduction analysis correlating the expression of HERVs with the LUAD and LUSC molecular subtypes (new figures 1C and 1D). This analysis revealed three main clusters in LUADs, where tumor samples belong to distinct LUAD molecular subtypes (Figure 1C). In LUSCs, we obtained two major clusters; one shows tumor samples belonging to the molecular subtypes SQ.1, while in the other, the molecular subtypes SQ.2a and SQ.2b were predominant (Figure 1D).

Moreover, we performed an additional differentially expression analysis (new supplementary file S5) in which we compare the early stages of NSCLC (IA) vs. the advanced stages (III and IV). Through the analysis we identified >170 dysregulated HERVs in LUAD, while in LUSC, no strong significant results were detected. These findings suggest that the tumor stages do not necessarily reflect a dysregulation HERVs.

Following the Reviewer's suggestion, we employed a cohort-specific correlation networks analysis showing the connections between the identified differentially expressed HERVs and a list of selected oncogenes and adaptive immunity regulators used in the clinic as targets for NSCLC treatment (new figures 5A and 5B and new supplementary files S8 and S9). This analysis showed several HERVs to be significatively (the vast majority positively) correlated with oncogenes and adaptive immunity regulators in both LUAD and LUSC cohorts.

We also appreciated the Reviewer's suggestion about analyzing POPLAR and OAK immunotherapy studies, and we understand the importance of profiling such datasets. However, we believe this suggestion is beyond the scope of our study. In fact, as stated along with the manuscript, we mainly aimed at profiling through a descriptive approach, for the first time, HERV expression with a locus-specific resolution using an established methodology to identify novel molecules dysregulated in NSCLC; also evaluating their potential roles in regulating signaling pathways by employing in-silico models and their association with patient survivals. Notwithstanding, as observed by the Reviewer, given the importance of immunotherapy in NSCLC and the relevance of identifying novel biomarkers related to the response of immune checkpoint inhibitors, we believe that such a topic must be addressed with a dedicated study. Indeed, in the future, we plan to profile HERVs with the same methodology in these two resources with this scope. In the revised version of the manuscript, we have also discussed this aspect in the Discussion section mentioning that such a study would be necessary as a future perspective.

Reviewer 3 Report

The manuscript entitled "Transcriptome analysis of Human Endogenous Retroviruses at locus-specific resolution in Non-Small Cell Lung Cancer" reports a bioinformatics analysis pipeline and relevant findings from analyzing public databases. The manuscript was well-written and well-organized, and the findings have the potential to help us further understand HERVs in cancer.

The results presented by the authors generally support their main conclusions. However, before this manuscript can be recommended by this reviewer for acceptance, the authors will need to address the following points:

1. Locus-specific resolution was claimed as the main novelty of this study. However, have the authors compared the locus-specific analyses with general analyses without paying attention to genomic loci, as other papers did? Is the locus-specific analysis pipeline equipped to reveal more useful information? The authors did not describe this and didn't explain why they didn't do that. We can not just assume that locus-specific analysis will be definitely better.

2. Since locus-specificity is the main novelty of this study, have the authors tried to first introduce some integration of HERVs artificially and then do RNA-seq and identify the artificial integration using their newly developed pipeline?

3. Have the authors done any functional validation for their findings, say testing changes in the expression those potential genes that may be regulated by HERVs in cancer cells?

4. The authors claimed that only very few related papers have been published on the topic. However, several major types of cancer have been analyzed for HERVs. Have the authors compared their findings with those reports? Similarities? Major differences? It seems that, at least to this reviewer, the authors just changed the cancer type to lung cancer but used almost the same types of analyses for HERVs. Anything new besides the claimed "locus-specificity"?

Author Response

Reviewer's commentThe manuscript entitled "Transcriptome analysis of Human Endogenous Retroviruses at locus-specific resolution in Non-Small Cell Lung Cancer" reports a bioinformatics analysis pipeline and relevant findings from analyzing public databases. The manuscript was well-written and well-organized, and the findings have the potential to help us further understand HERVs in cancer.

The results presented by the authors generally support their main conclusions. However, before this manuscript can be recommended by this reviewer for acceptance, the authors will need to address the following points:

Authors' response: We appreciate the time and effort that the Reviewer dedicated to providing feedback on our manuscript.

Reviewer's comment: 1) Locus-specific resolution was claimed as the main novelty of this study. However, have the authors compared the locus-specific analyses with general analyses without paying attention to genomic loci, as other papers did?

Is the locus-specific analysis pipeline equipped to reveal more useful information?

The authors did not describe this and didn't explain why they didn't do that. We can not just assume that locus-specific analysis will be definitely better.

Authors' response: We thank the Reviewer for allowing us the opportunity to address this point. We want to clarify that, in this paper, we did not develop our own pipeline for analyzing HERVs at locus-specific from RNA-Seq data. In fact, we employed an already established method, called Telescope (PMID: 31568525), for quantifying these molecules aiming at investigating, for the first time, those dysregulated in NSCLC and their impact on signaling pathways and patients' survival. We better highlight this through the paper as the main novelty of our study since no other similar reports that used the same methodology have been reported on NSCLC.

The pipeline we used in our study, Telescope, has already been tested and compared with other methodologies proving its accuracy in quantifying such elements (PMID: 31568525). Moreover, it has been successfully used in different tumor types, as cited in the Introduction section (PMID: 32872377; PMID: 33941616; PMID: 35349481). Therefore, it is not our scope to prove the efficacy of this methodology, which has already been proven.

The importance of analyzing HERVs with a genomic locus-specific resolution is due to the repetitive nature of such elements. In fact, as the Reviewer correctly said, other studies have analyzed HERVs for different tumor types using non-locus specific analysis pipelines. The pipelines used in such studies were not explicitly designed for HERV characterization and quantification; therefore, they could not quantify specific HERVs, providing only a summarized expression of the HERV families without discriminating the single elements. Indeed, quantifying HERVs with standard short-read RNA-Seq technologies is challenging due to the repetitive nature of such elements and the consequent uncertainty in fragment assignment because of sequence similarity. Therefore, specific pipelines such as Telescope have to be used to quantify specific HERV elements rather than just the families. Using an analogy between HERVs and coding-protein genes, we suppose that a researcher would prefer to know which specific genes are dysregulated in a specific condition rather than just know which family is affected without discriminating against the actual genes. For this reason, we deem that a locus-specific analysis able to quantify the specific HERVs is better than an analysis that lacks this resolution.

Reviewer's comment: 2) Since locus-specificity is the main novelty of this study, have the authors tried to first introduce some integration of HERVs artificially and then do RNA-seq and identify the artificial integration using their newly developed pipeline?

Authors' response: We thank the reviewer for allowing us the opportunity to clarify this aspect. As we wrote in the previous comment, the HERV locus-specific analysis is not the main novelty of this study. In fact, by using this methodology, we were able to identify, for the first time, the dysregulated HERVs in LUAD and LUSC, their potential impact on signaling pathways, and their association with patient survival. To the best of our knowledge, no other similar studies have been reported on NSCLC.

Moreover, we want to clarify again that we did not develop a new pipeline in our study. We used Telescope as an established method, published in 2019 (PMID: 31568525), and successfully used in other studies to quantify HERVs from standard short-reads RNA-Seq data (PMID: 32872377; PMID: 33941616; PMID: 35349481). The authors of this methodology have already compared their tool against other methodologies proving its efficacy in addressing its task (PMID: 31568525). Therefore, it is not within the scope of our study to benchmark the ability of Telescope to quantify HERVs accurately. Our study aimed to generate new knowledge about the importance of HERVs in NSCLC.

Reviewer's comment: 3) Have the authors done any functional validation for their findings, say testing changes in the expression those potential genes that may be regulated by HERVs in cancer cells?

Authors' response: We thank the reviewer for this valuable question. We are aware of the limitations of our study that the lack of experimental validations of our findings in an external cohort. However, the goal of our study was to use bioinformatics methodologies and a valuable repository, such as TCGA, to study the involvement of HERVs in several aspects of NSCLC biology. More precisely, this paper identifies the HERVs dysregulated in both LUAD and LUSC, their impact on signaling pathways using in-silico predictions, and their association with patient survival. In addition, in the revised version of the manuscript, we performed a new UMAP reduction analysis correlating the expression of HERVs with the LUAD and LUSC molecular subtypes (new figures 1C and 1D). Moreover, we performed an additional analysis (new supplementary file S5) comparing the early stages of NSCLC (IA) against the advanced stages (III and IV). We also made cohort-specific correlation networks showing the connections between the identified differentially expressed HERVs and a list of selected oncogenes and adaptive immunity regulators used in the clinic as targets for NSCLC treatment (new figures 5A and 5B and new supplementary files S8 and S9). Although future validations are needed, we believe this study might be the first step toward identifying HERVs that play a pivotal role in NSCLC pathogenesis.

Reviewer's comment: 4) The authors claimed that only very few related papers have been published on the topic. However, several major types of cancer have been analyzed for HERVs.

Have the authors compared their findings with those reports? Similarities? Major differences?

It seems that, at least to this reviewer, the authors just changed the cancer type to lung cancer but used almost the same types of analyses for HERVs. Anything new besides the claimed "locus-specificity"?

Authors' response: We thank the Reviewer for this comment. We are aware that several papers have been published about the roles of HERVs in cancer. However, as we reported in the Introduction section, very few studies have used bioinformatics pipelines able to quantify specific HERV elements rather than families from RNA-Seq data making it more difficult to compare our findings with those ones reported by other authors. Indeed, in most of these studies, HERVs have been quantified at the family level without discriminating against the single elements included in these families because of their sequence similarities.

Moreover, the lack of a unique classification of these elements makes it challenging to compare the results obtained from studies that have used different methodologies for their quantification and nomenclature with ours. This aspect is also another reason why a locus-specific analysis that gives the information about which specific HERV with its genomic coordinates is deregulated removes the problem of ambiguity, giving the opportunity uniquely identify the HERVs. Nevertheless, even in the previous version of our manuscript, we deeply analyzed the literature looking for other studies that have reported the dysregulation of HERVs in lung cancer. Some of them were cited in the manuscript, and a description of their findings compared with ours was done. For example, our study found several HERV families to be dysregulated in both LUAD and LUSC. Families with a higher percentage of differentially expressed HERVs included HERV-H, HERV-K, and MER4. Interestingly, the up-regulation of the HERV-H and HERV-K families has also been observed by other authors in NSCLC, and they have been proposed as potential novel diagnostic biomarkers (PMID: 29274460; PMID: 35359739; PMID: 31875158). Moreover, in another study, the up-regulation of the MER4 family has been linked to Progression-Free Survival (PFS) and Overall Survival (OS), and they have also been proposed as novel biomarkers for the response to Immune Checkpoint Inhibitors (ICI), which, currently, are becoming the standard first-line treatment of advanced NSCLC for patients with high PD-L1 expression (PMID: 35277462).

Ultimately, we also want to clarify that we did not just perform the same analysis that other authors did on different cancer types changing only the tumor type. Besides the novelty of doing this study for the first time on NSCLC and the locus-specificity of our analysis, which we believe is extremely important for the reasons explained in the previous comments, several bioinformatics analyses have been performed in this study that have never been collectively performed in other studies, even with different tumor types. Examples include the clustering analyses with histological and molecular NSCLC subtypes, HERV-specific pathway analyses based on neural network predictions, correlation networks between HERVs and relevant oncogenes and adaptive immunity regulators used in the clinic for lung cancer treatment, and association of HERV expression with NSCLC patients’ survival.

Reviewer 4 Report

In this report the Authors examine the expression of human endogenous retroviruses in non-small cell lung cancer, both adenocarcinoma and squamous cell subtypes, in the Cancer Genome Atlas database, by a technique that allows high resolution at locus-specific level. HERV expression may be relevant to pathogenesis, prognosis, early detection and treatment of human cancer, and this is, to my knowledge, the first report to analyze their differential expression in lung cancer subtypes with this methodology. While I do not have the expertise to critically assess the informatics methodology they have utilized, I deeply appreciated the general design of the study and the results reported. I only have a general comment on the Methods: it would be helpful to correlate the HERV expression not only to histological subtypes, but also to molecular subtypes and to molecular alterations commonly exploited in the clinic (e.g., EGFR mutations).

I do not have specific comments.

Author Response

Reviewer's comment: In this report the Authors examine the expression of human endogenous retroviruses in non-small cell lung cancer, both adenocarcinoma and squamous cell subtypes, in the Cancer Genome Atlas database, by a technique that allows high resolution at locus-specific level. HERV expression may be relevant to pathogenesis, prognosis, early detection and treatment of human cancer, and this is, to my knowledge, the first report to analyze their differential expression in lung cancer subtypes with this methodology. While I do not have the expertise to critically assess the informatics methodology they have utilized, I deeply appreciated the general design of the study and the results reported.

Authors' response: We appreciate the time and effort that the Reviewer dedicated to providing feedback on our manuscript. The insightful comments are valuable to improving our manuscript. 

Reviewer's comment: I only have a general comment on the Methods: it would be helpful to correlate the HERV expression not only to histological subtypes, but also to molecular subtypes and to molecular alterations commonly exploited in the clinic (e.g., EGFR mutations).

I do not have specific comments.

Authors' response: Thanks for this valuable comment. Following the Reviewer's suggestion, in the revised version of our manuscript, we performed a new UMAP reduction analysis correlating the expression of HERVs with the LUAD and LUSC molecular subtypes (new figures 1C and 1D). This analysis showed three main clusters in LUADs, where tumor samples belong to distinct LUAD molecular subtypes (Figure 1C). Moreover, in LUSCs, we obtained two major clusters; one shows tumor samples belonging to the molecular subtypes SQ.1, while in the other, the molecular subtypes SQ.2a and SQ.2b were predominant (Figure 1D).

In addition to that, we performed a new analysis (new supplementary file S5) comparing the early stages of NSCLC (IA) against the advanced stages (III and IV). Finally, we also performed a cohort-specific correlation networks analysis showing the connections between the identified differentially expressed HERVs and a list of selected oncogenes and immunity regulators used in the clinic as targets for NSCLC treatment (new figures 5A and 5B and new supplementary files S8 and S9).

Round 2

Reviewer 2 Report

Unfortunately, the authors have not provided any new analyses with conclusions that would be of interest to lung cancer community. 

Reviewer 3 Report

The authors provided good answers to the questions raised by this reviewer. However, the authors did put "locus-specific resolution" in the title of this manuscript, it is not very convincing to say in the response letter that locus-specificity is not a major novelty.

As said above, the authors clarified some of the reviewer's concerns, but what is still missing about the solid experimental data continues to weaken this study. Therefore, this reviewer suggests that the authors perform relevant experiments and provide data from at least one of the following comments in the original review:

"2". Since locus-specificity is the main novelty of this study, have the authors tried to first introduce some integration of HERVs artificially and then do RNA-seq and identify the artificial integration using their newly developed pipeline?

"3". Have the authors done any functional validation for their findings, say testing changes in the expression those potential genes that may be regulated by HERVs in cancer cells?

If web-lab experiments are too challenging to the authors, this reviewer would recommend that they submit it to a bioinformatics-focused journal.